# Special Cases of Using Visualization Technology for Analyzing the Dynamics of Gaseous Environment

**Mikhail Sotskiy \*, Denis Levin and Victor Selivanov**

Faculty Special Mechanical Engineering, Department High-Precision Airborne Devices, Bauman Moscow State Technical University, 105005 Moscow, Russia; dlenine@yandex.ru (D.L.); vicsel@list.ru (V.S.)
**\*** Correspondence: msotsky.bmstu@mail.ru; Tel.: +7-903-120-8077

**Abstract:** A new visualization technology is presented, which was used in applied research when observing and modeling the dynamics of the flow of gaseous environments. In the process of developing and improving the technology, a set of experimental results was compiled to study the phenomenon of combustion and detonation of a hydrogen-oxygen mixture, as well as the phenomena of propagation, action, and interaction of shock waves and gas-dynamic structures. On the example of analyzing data on the dynamics of the formation of a vortex ring, the possibilities of verifying the computational model of the implemented physical process are shown. The presented results reflect the level of information content when using technology to carry out tests.

**Keywords:** gas mixture; detonation; shock wave; imaging technology; vortex ring; verification

## 1. Introduction

The experimental–theoretical study of the dynamics and kinetics of reactive rheological media (gas, liquid, mixture) is acquiring new relevance with the development of ways to use these media in innovative pulsed and energy devices. Particular attention is paid to the issues of dynamics of gaseous media in channels and cavities of devices [1–3], flow turbulization, temperature and energy thresholds of transitions of media to ignition, detonation, and steady-state propagation velocities. The work [4] can be cited as an example of the study of these issues based on a thorough reconstruction of a virtual picture of gas dynamics processes. The interaction of numerical and experimental visualization for practical use has been considered in many investigations [5–11]. An example of the technical use of visualization of gas detonation in channels for verification of computational models of processes can be found in the research [12]. The experimentally recorded time sweep of the flame front propagation in a stoichiometric hydrogen–oxygen mixture along the axis of the transparent tube served as the basis. By the slope of the tangent to the trajectory of the combustion front, the velocity of the detonation wave V was determined. This measuring technology is traditional; it provides reliable data on the rate of the process but does not reveal the physical picture in its entirety.

The aim of this work is to demonstrate the possibilities of revealing the phenomenology of physical phenomena from the analysis of the results of experiments on innovative visualization technology [13]. A series of experiments in the field of ballistics is implemented in a given range of initial data. Based on the results of processing the obtained visual data, quantitative predictions are made based on the laws of known theories. This is in contrast to setting up experiments in the scientific method. The purpose of an experiment with a scientific approach is to test a scientific hypothesis. Phenomenology is concerned with the philosophical notion that these predictions describe expected behaviors for phenomena in reality. These models can reveal the phenomenology of the investigated phenomena realized in the investigated range of the initial conditions of the series of experiments.

We chose different variants of fast-flowing gas-dynamic processes for demonstration. Representation of optical recordings in the form of a series of characteristic frames cannot reflect the dynamics, stages, and wave essence of the phenomena. The phenomenology of the phenomenon is revealed only by direct viewing of video files. Below are some typical results from a series of experiments carried out on a research facility when using the method for carrying out ballistic experiments according to the author's patent RU 2625404, 2017, Bul. No. 20. The physical phenomenon of the formation and dynamics of propagation of a vortex ring has been studied in more detail, acquiring new, modern applications [14]. The phenomenon has a long history of modeling and practical use. In this paper, the phenomenon is considered on the basis and taking into account the results of well-known scientific and practical studies [15–17] and using optical imaging technology.

## 2. Optical Imaging Technology and Equipment

The device and method for carrying out ballistic experiments were developed and applied in laboratory experiments to visualize various ballistic processes. A research launcher was created using the device (RU 2619501, 2017, Bul. No. 14). The active unit of the installation with a guide element and a supply line are shown in Figure 1a.

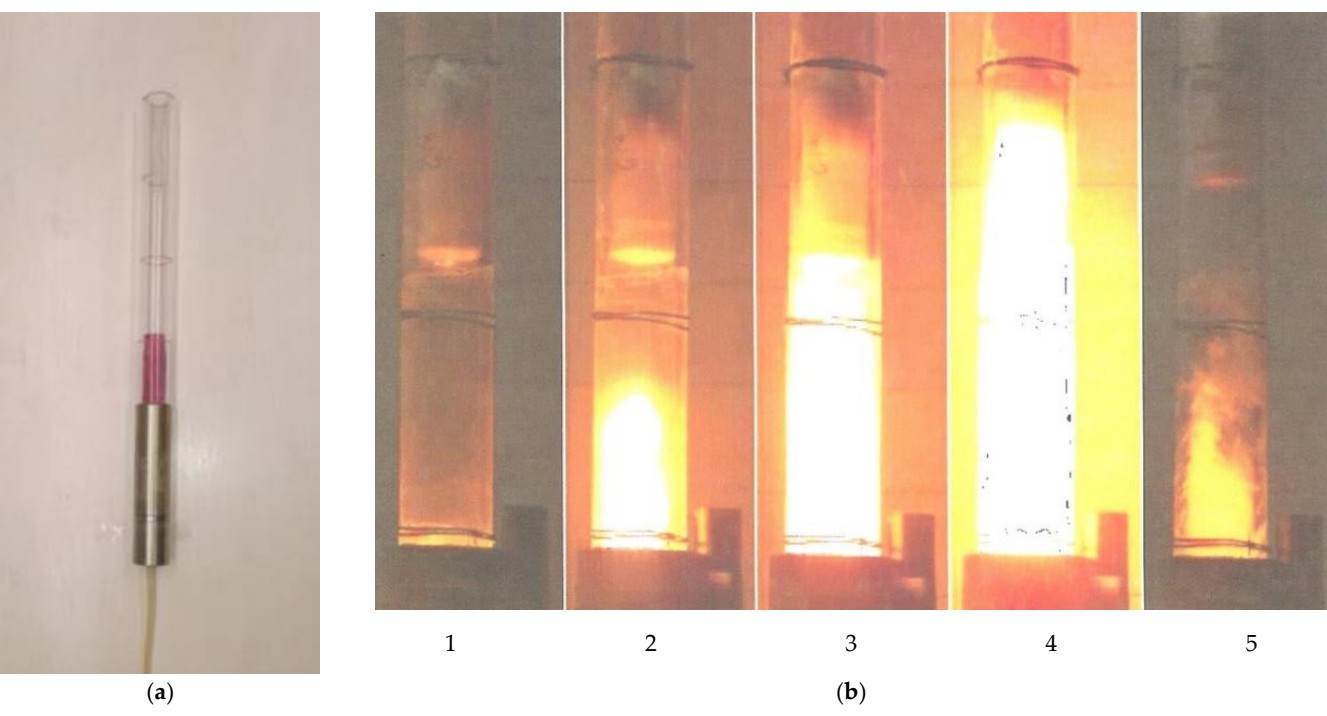

**(a)**              1        2        3        4        5        **(b)**

**Figure 1.** The active unit of the setup (**a**) and the initial phase of the process (**b**) of the effect of the detonation products of the hydrogen–oxygen mixture on the probe during its acceleration in the guiding element.

The active unit of the installation contained a guiding element, the object under investigation (in the presented case, a probe) or a rheological medium, and a control cavity formed by a closed elastic shell. The control cavity was located in the starting zone of the guiding element and filled with a working gas. The control cavity was also connected through a pipeline and shut-off-regulating equipment to a working gas source made in the form of an electrolyzer. Locking and regulating equipment contained a spark gap. The operation of the installation was provided by the control equipment. The mixture was formed as a result of the electrolysis of water and fills the control cavity, made in the form of an elastic destructible shell of the required volume (highlighted in red in Figure 1a). The low optical density of the mixture detonation products made it possible to use a guiding element made of an optically transparent material in the active unit of the launcher. The active unit was located in an armored chamber. The process was illuminated

through the windows. Optical registration was performed with a high-speed video camera Phantom v1610.

The operation of the installation and the implementation of the method were based on the conversion of the energy of the working gas—a mixture of oxygen and hydrogen. The installation operation is described in detail, for example, in [4]. The shooting frequency in a series of experiments was from 14,000 to 2,000,000 frames/s. The working gas was a mixture of hydrogen and oxygen in a stoichiometric volume ratio of 2 to 1.

The data in Figure 1b are shown for a guide element 1000 mm long with an outer diameter of 40 mm and an inner diameter of 30 mm. Frames 1 and 2 captured the moment of movement of the detonation wave front up to the bottom of the stationary probe. The middle frame 3 in Figure 1b captured the position of the bottom part at the moment of the fourth reflection of the shock wave from it, while frames 4 and 5 captured the position of the probe in the period of the accelerated movement after repeated impacts.

The realized physical phenomena have an essentially wave nature. For this reason, the phenomenology of phenomena is revealed most clearly when viewing and analyzing the primary file saved in the Cine format. This is one of the possible (Cine, Cine Compressed, Cine RAW, AVI, etc.) saving formats, containing the most complete amount of information about the observed process. The sequence of frames presented in the figures below only reflects the characteristic stages of the studied physical phenomena.

It is possible to evaluate the energetics of the processes by analyzing the data of a series of experiments that implement the processes with different initial parameters. The experiments were carried out in the range of variation of the length $L$ of the guide element from 500 to 1000 mm, the volume $U$ of the control cavity with a working gas from 100 to 400 mL, and the mass $m$ of the probe from 70 to 150 g. The series of photographs presented in this work are given only for illustration of research technology. Quantitative measurements of the kinematic characteristics of the observed process are realized directly when processing a video file saved in the Cine format. The set of the initial parameters of the experiment ($L$, $U$, and $m$) determines the values of the parameters realized in a particular ballistic process. These are the initial velocity $V$ of the probe, the average value of the acceleration $dV/dt$ of the probe, and the muzzle energy $E$ of the probe. As a result of processing the experimental data, the range of variation of these parameters for a guide element with an inner diameter of 30 mm was determined (summarized in Table 1).

**Table 1.** The range of variation of unit parameters.

| $V$, m/s | $dV/dt$, m/s$^2$ | $E$, J |
| --- | --- | --- |
| 20.01–52.25 | 610–4282 | 8–96 |

The results obtained can be taken into account when preparing experiments with other initial conditions for experiments in a series. The values of the motion parameters of the inertial measuring probe recorded in the experiments were used to verify the calculation models of the processes implemented in the experiments.

## 3. Results and Discussion

This section presents a set of results of experiments carried out to study the phenomenon of detonation of a hydrogen–oxygen mixture. The set of visualization of particular options under consideration was compiled in the process of developing and improving the technology in the study of the phenomena of propagation, action, and interaction of shock waves and gas-dynamic structures. Video recordings of the processes under consideration were obtained using a high-speed video camera Phantom v1610.

The objectives of this particular version of the study are as follows:

(1)   Study of the mechanisms of transition from combustion to detonation under different initial conditions of the experiment;

(2) Experimental study of the transfer of detonation from an active charge consisting of a gas mixture enclosed in an elastic shell to a passive charge, also consisting of a gas mixture enclosed in an elastic shell through the air and through a liquid medium (water).

### 3.1. Propagation of Detonation of the Mixture in Curved Channels

In the series of experiments presented below, the initiation of the active volume of the hydrogen–oxygen mixture was carried out by an electric discharge in a special initiating device. The initiating device, connected to the active charge by a flexible plastic tube, allows using a system of valves to accumulate the gas mixture in the ignition chamber of the high-voltage spark plug, flexible plastic tube, and active volume. At the right time, the mixture was ignited, and the impulse was transmitted to the active volume. The transfer process was investigated in a specially conducted series of experiments with different diameters of the plastic tube: 4, 6, 8, 10, and 12 mm. The tubes were formed into a spiral ring and fixed on a plane. The movement of the detonating pulse through the tubes to the active volume was recorded by a video recorder. Figure 2 shows the results of one of the experiments from the implemented series.

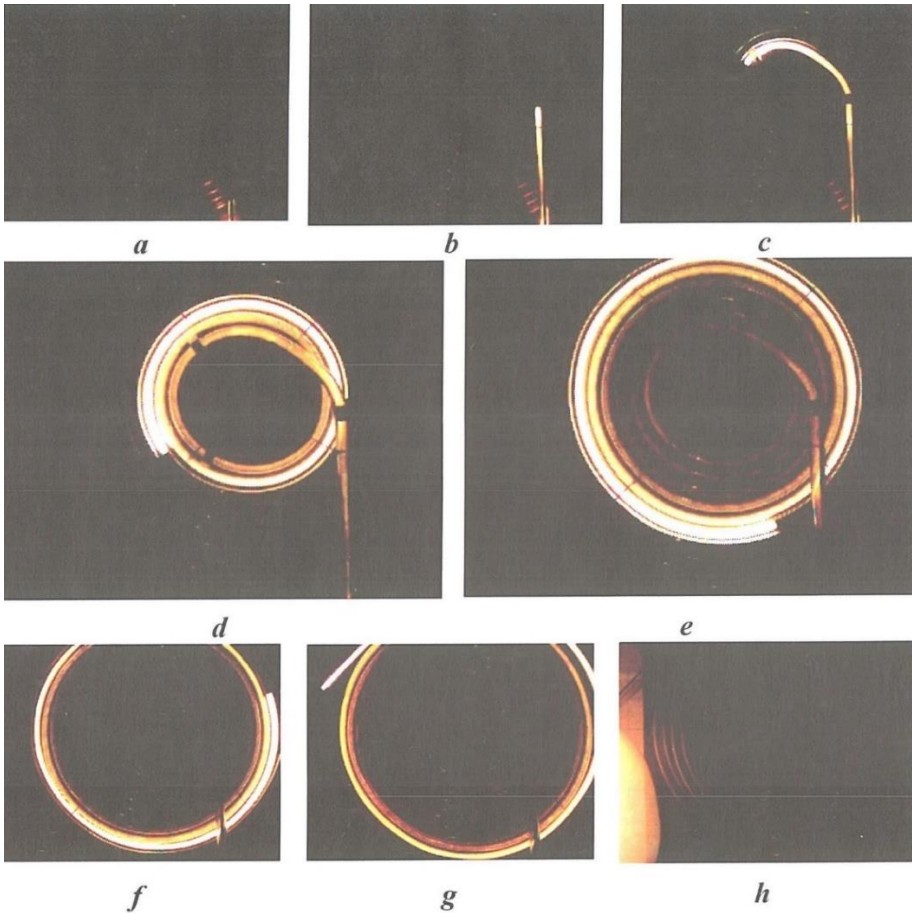

**Figure 2.** Separate video recordings of the detonation front movement along the annular tube to the working volume of the hydrogen–oxygen mixture.

On frame *h* (Figure 2), an expanding elastic shell of the active volume of the hydrogen–oxygen mixture is visible in the lower-left corner of the frame. Measurements have established a small dependence of the pulse transfer rate on the tube diameter in the investigated range (about 2900 m/s).

The series of experiments carried out demonstrates the reliable transmission of the detonating pulse through the plastic tube to the active volume of the research facility.

### 3.2. The Transfer of the Detonation of the Mixture through the Air Gap

The active unit is mounted vertically on a special tripod and placed in an armored chamber. The guide element, in this case, is a shock tube 1000 mm long, 40 mm in outer diameter, and 30 mm in inner diameter, made of optically transparent material. The lower part of the guide element is placed in a metal sleeve. The sleeve performs the function of fixing it in the tripod and preventing the shock tube from breaking during the initiation of the active volume of the hydrogen–oxygen mixture. The active volume was an elastic envelope filled with a gas mixture and placed in the lower part of the shock tube. The passive charge was also an elastic shell filled with a gas mixture, placed in the upper part of the shock tube at the required distance from the active charge. According to the experimental conditions, the gap between the charges was either filled with air or filled with a liquid (water). The installation was located in an explosion chamber, and illumination and video recording of the processes with a high-speed video camera Phantom v1610 were carried out through the windows in an armored chamber. The video shooting speed was 84,000 frames/s.

In the experiments, a hydrogen–oxygen mixture was used in a ratio close to stoichiometric. To achieve a constant composition of the gas mixture, the system was purged with a hydrogen–oxygen mixture for at least 5 min.

Figure 3 shows the results of registration of the process with the following initial data: the length of the active volume L = 350 mm; passive volume L = 415 mm; air gap L = 230 mm.

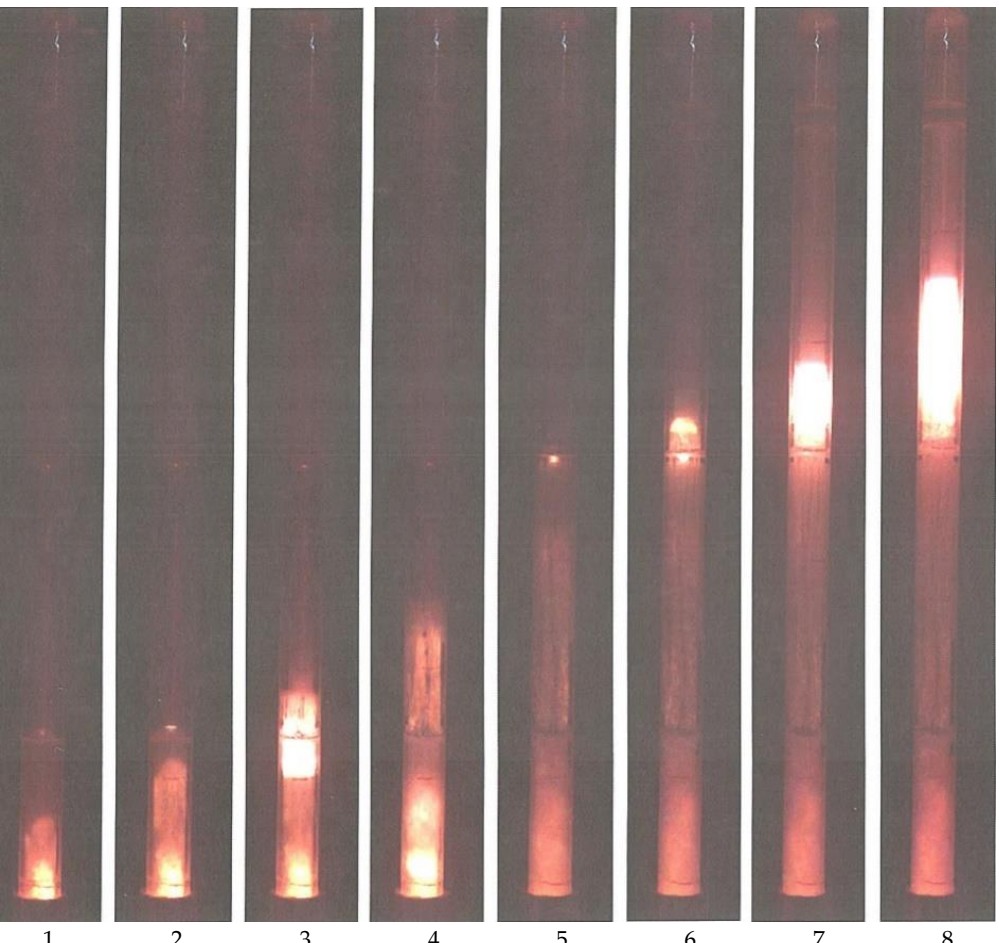

**Figure 3.** Separate video recordings of the detonation transfer process through an air gap of 230 mm from the lower to the upper volume of the hydrogen–oxygen mixture (active volume 350 mm long).

The video recording frames show the following: The combustion front moves (frames 1, 2) along the active volume (up the pipe) at a speed of 850 m/s, then, due to gas compression at the upper boundary, a reflected shock wave is formed (frame 3), which propagates down the pipe at a speed of 1470 m/s and gradually decays. After the destruction of the shell of the active volume (frame 5), a shock wave runs upward through the pipe at a speed of 1340 m/s together with the detonation products of the active charge. The passive volume is initiated in the detonation mode (frames 6–8), the wave propagates at a speed of 2846 m/s.

Figure 4 shows the results of registration of the process, with the following initial data: the length of the active volume is 400 mm; passive, 320 mm; air gap, 237 mm.

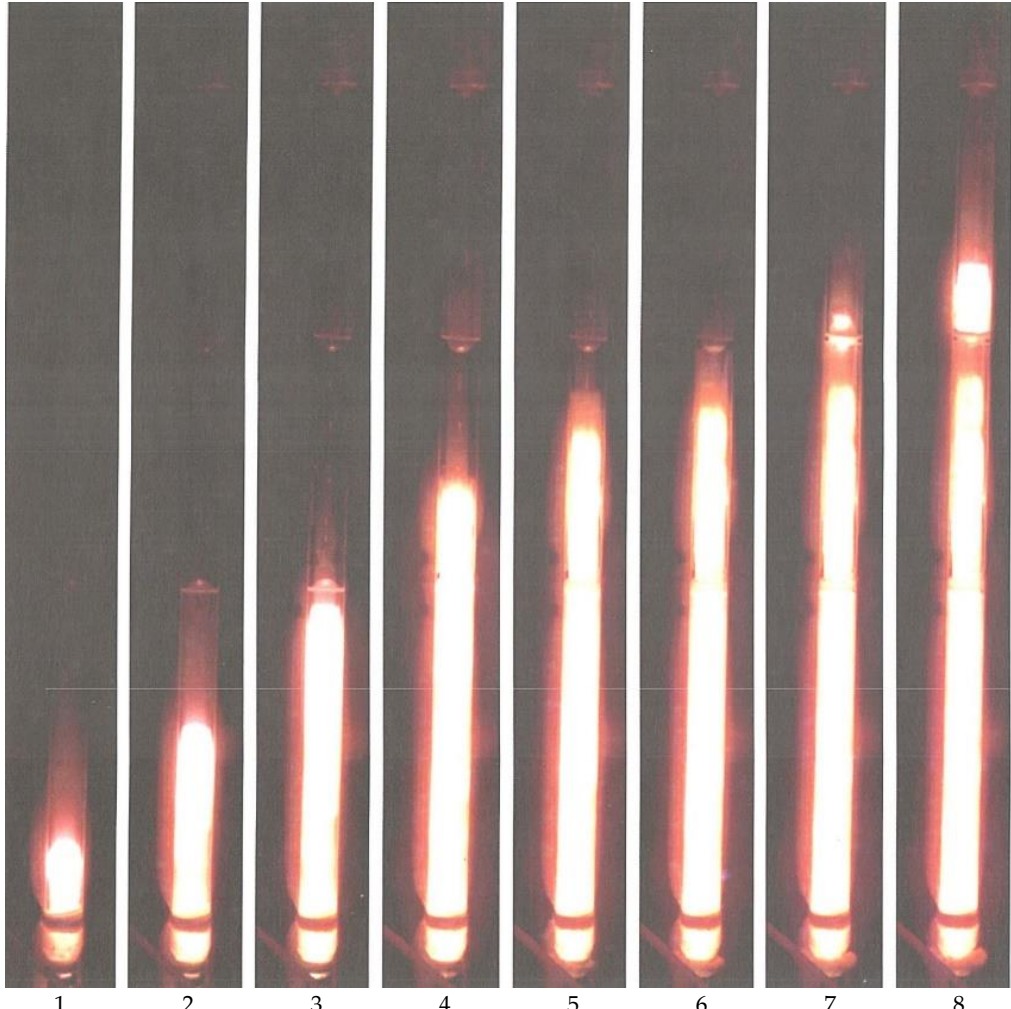

| 1 | 2 | 3 | 4 | 5 | 6 | 7 | 8 |

**Figure 4.** Selected video recordings of the detonation transmission process through an air gap of 237 mm from the lower to the upper volume of the hydrogen–oxygen mixture (active volume 400 mm long).

The following is observed on the video recording: the initiation of the active volume immediately occurs in the detonation mode, the wave propagates up the shock tube at a speed of 2753 m/s. At the moment of reaching the upper boundary (frame 3), the contact discontinuity decays—the shock wave goes upward (frames 4–6) together with the detonation products (the wave velocity is 1270 m/s), and the reflected shock wave goes downward. Further, detonation is excited in the passive volume and the detonation propagates (frames 7–8) upward to the entire passive volume (the detonation wave velocity is 2925 m/s).

Thus, in the second case, we also observe the transfer of detonation from the active volume to the passive volume through the gap (air layer) by means of adiabatic compression in the shock wave.

### 3.3. Transfer of the Detonation of the Mixture through the Water Gap

To create a clear boundary between charges and liquid, a technique was developed for supplying gas to a passive volume with the simultaneous use of a drainage tube to evacuate the air in the shock tube.

Figure 5 shows the results of registration of the process with the following initial data: the length of the active volume of the mixture is 400 mm; passive, 500 mm; the length of the gap filled with liquid (water) is 10 mm.

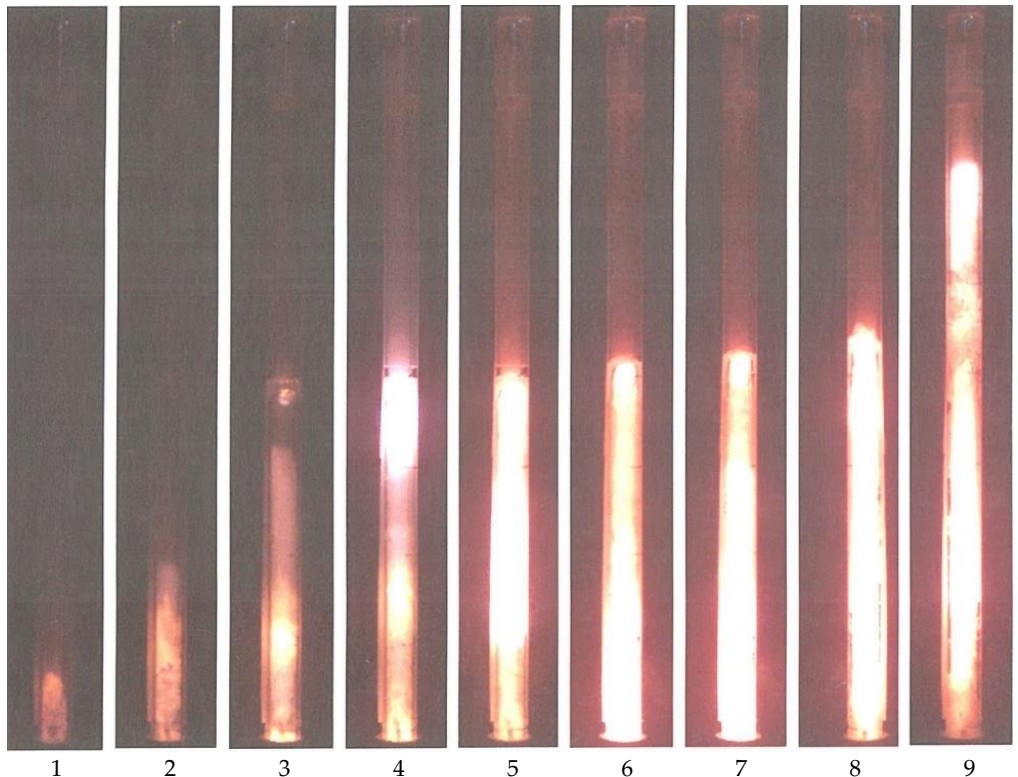

**Figure 5.** Selected video recordings of the detonation transfer process through a 10 mm water gap from the lower to the upper volume of the hydrogen–oxygen mixture (active volume 400 mm long).

On video recording, the following is observed: the initiation of the active volume occurs in the combustion mode, the wave propagates up the tube at a speed of 708 m/s (frames 1–3), due to gas compression, a detonation center is formed at the upper boundary of the active volume (frame 4). Then, the detonation wave propagates down the tube to the entire active volume at a speed of 2216 m/s. At the same time, the elastic shell begins to move along with the water up the pipe (frames 5–7) and, similar to a piston, presses the shell of the passive volume. After the detonation wave in the active charge reaches the lower end, it is reflected, and the reflected shock wave begins to propagate upward with a speed of about 1500 m/s. Next, the passive volume is initiated in the detonation mode (frames 8, 9), and the wave propagates upward through the volume.

The results of a series of experiments with the transfer of detonation through the gap are used to establish the features of the realized physical phenomena and to test the numerical calculations of the investigated processes.

### 3.4. Detonating Pulse Output to the Water Surface

Additionally, in the macro mode, the physical phenomenon of cavitation was investigated when a detonating pulse exits the water surface (Figure 6).

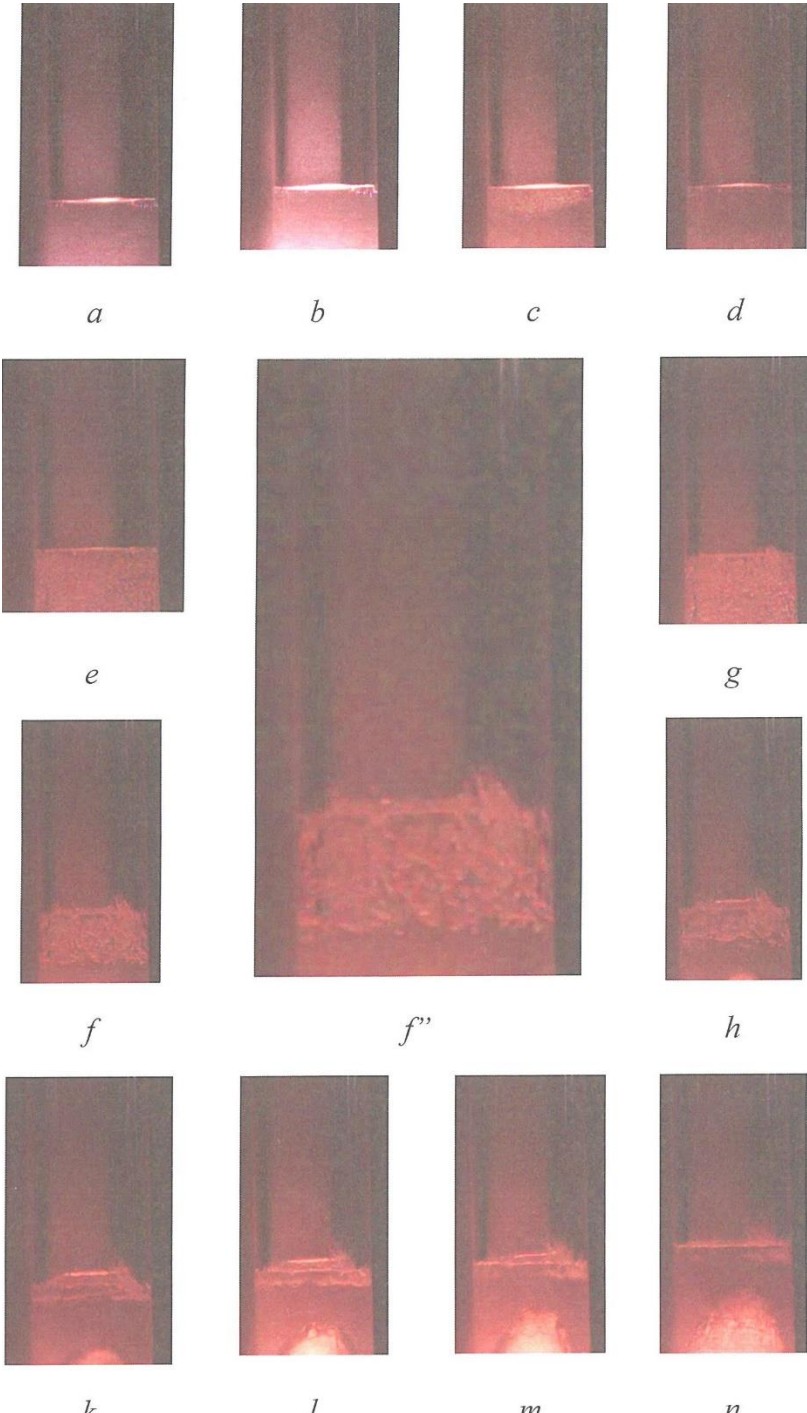

**Figure 6.** Separate video recording frames of cavitation on the surface of the water gap during the period of the shock wave emerging from the detonation of the hydrogen–oxygen mixture.

Frame demonstrates in macroscale the moment when the cavitating wavefront reaches the water surface.

Below are some representative results from a series of experiments carried out on a research launcher using the method of accelerating a body in a ballistic experiment.

*3.5. Vortex Ring Generation*

3.5.1. Phenomena Physical Modeling

To experimentally determine the speed of the vortex ring (VR), a high-speed video recording was performed (the time between frames is 50 ms, the frame exposure is 5 μs) with the Phantom v1610 camera. To visualize the vortex rings, smoke was previously inserted into the volume of the guide element (in this case, also the shock tube) above the active volume of the hydrogen–oxygen mixture. The active volume was initiated, and a detailed physical picture of the formation and propagation of the vortex ring was recorded. The registration variant is shown in Figure 7.

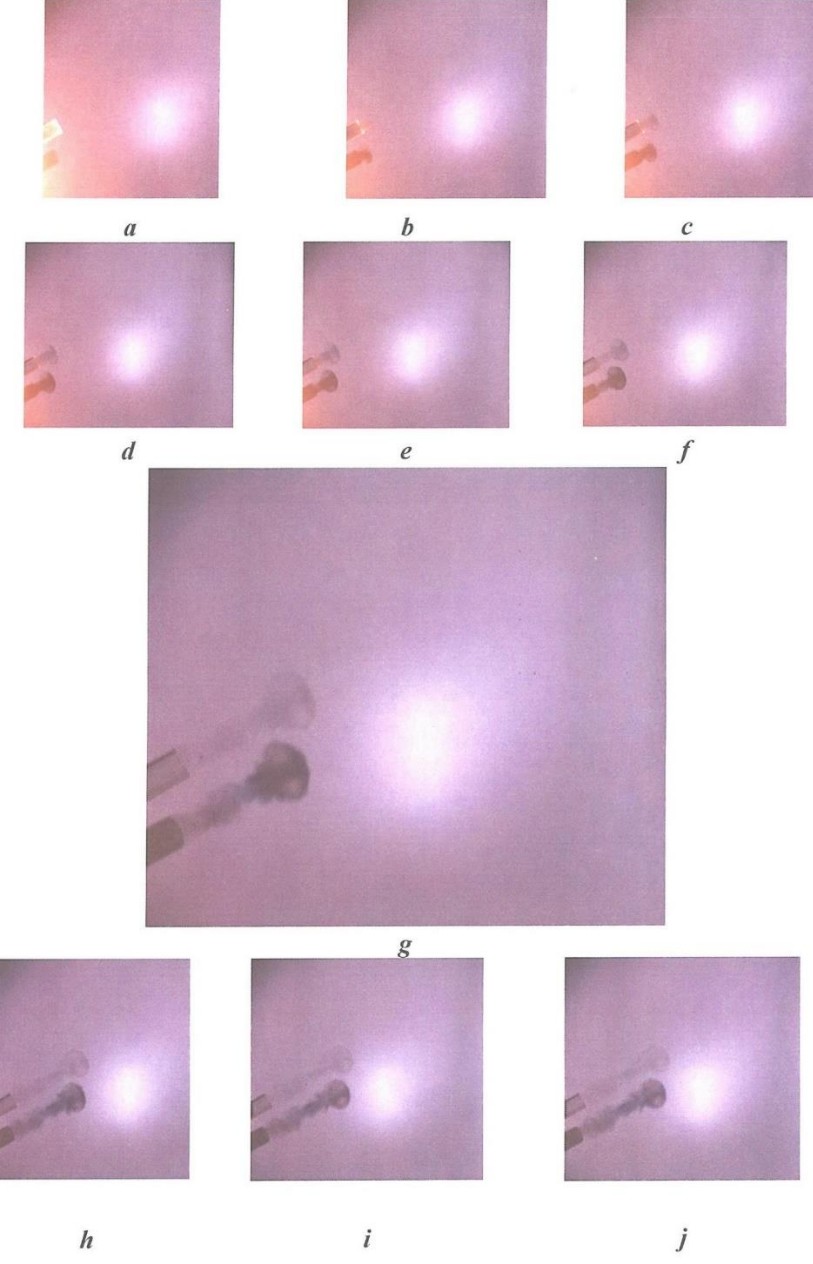

**Figure 7.** Separate video recording frames of the process of forming a gas ring after the detonation of a hydrogen–oxygen mixture in the guide element.

To improve the visual display of the rings, high-intensity illumination and a white substrate were used, so that in the future, not the rings themselves but their shadow displays on the substrate were processed. The processing was performed in the Phantom

Camera Control Application program using the built-in tools for measuring the geometric characteristics of the image. The initial moment of time was taken as the moment of the final ring formation, which occurred 0.85 ms after the start of the exit of the colors' flow from the nozzle of the impact pipe at a distance from the cutoff of ≈1.3 of the nozzle diameter. Next, the distance traveled by the ring from the nozzle section of the shock tube was determined with a step of $\Delta t$ = 0.25 ms. The average speed of the ring over the traversed interval $\Delta l$ was determined as the ratio $\Delta l / \Delta t$.

The average coefficient of expansion of the ring $\alpha$ was estimated on the basis of 200 mm and was 0.043. The initial speed of the ring was 100 m/s.

### 3.5.2. Theoretical Representation of Vortex Ring Formation and Propagation

The vortex ring at the main stage of the propagation trajectory is a steady-state aerodynamic structure consisting of a toroidal vortex, inside which almost all the vorticity is concentrated, and the "atmosphere" of the vortex—the region surrounding the core and moving with it. The ring moves in the environment with the preservation of geometric proportions, the main of which is illustrated in Figure 8.

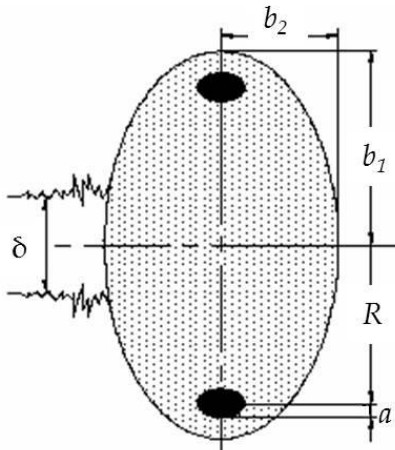

**Figure 8.** The main dimensions of the vortex ring: $D$ ($R$)—the outer diameter (radius) of the vortex—the distance between the centers of the vortex spirals; $a$—the radius of the vortex core; $b_1$ and $b_2$—the dimensions of the sides of the "atmosphere" of the vortex ring (if it is taken as an ellipsoid); $\delta$—the width of the vortex path.

Vortices are formed where the flow velocities change rapidly along the normal values to the fluid flow lines: in these places, large forces of viscous friction arise. In a viscous medium, there can be no sharp boundary between two flows, so there is a transition layer with rapidly changing velocities. The thickness of this layer is less, the lower the viscosity of the medium. Additionally, vortices are often formed when the velocity changes sharply in the direction that leads to the formation of interfaces. Here, too, there is a velocity gradient along the normal to the fluid flow lines and the formation of viscous friction forces that cause small gas particles to rotate first and then larger volumes. When the air is pushed out of the cavity, a boundary layer with significant vorticity is formed on the walls of the cavity nozzle due to viscous adhesion. The formation of the vortex begins when the jet front exits the nozzle into an undisturbed medium. The boundary layer under the action of the velocity field turns into a spiral. The leading edge of the mushroom head of the vortex is the boundary of the liquid that was in the pipe before the start of the jet outflow process. It follows from the results of the experiments that during the main time of the jet outflow, the velocity of the leading edge extension is approximately equal to half of the jet velocity. The jet that feeds the vortex retains its cylindrical shape even after exiting the hole, and only when entering the vortex does its cross section decrease. At high Reynolds numbers, the boundary layer is thin and practically represents a tangential discontinuity,

due to the instability of which (Kelvin–Helmholtz instability) the structure of the boundary layer is destroyed. The layers are mixed, which leads to the formation of a vortex core. It is worth noting that the core of the vortex forms only the front part of the boundary layer, and the rest of the latter enters the atmosphere of the vortex, where it spreads more or less evenly by turbulent pulsations and then gradually passes into the vortex path. Figure 9 shows the scheme of vortex ring formation.

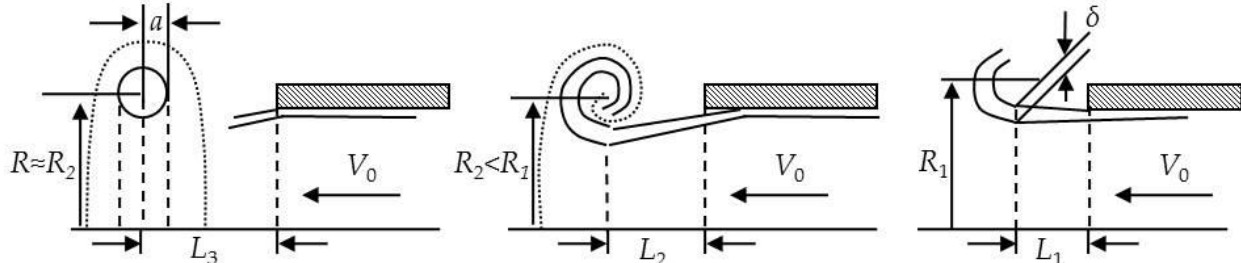

**Figure 9.** Diagram of the vortex ring formation process: δ —the thickness of the boundary layer; *R*—the radius of the VR (defined as the radius of a circle with zero velocities in the coordinate system associated with the VR); *a*—the radius of the core of the VR (the radius of the vortex thread).

It is experimentally defined that the core and the "atmosphere" of the vortex, respectively, account for 7.7% and 92.3% of the total mass of the medium carried by the vortex (at least after its formation), i.e., the mass of the core exceeds the mass of the "atmosphere" by about 12 times [14]. Thus, assuming $R/a = k$, where $k$ is the experimental coefficient, and the ratio of the volumes of the atmosphere and the core equal to 12 (the density of the vortex medium is determined by the density of the "atmosphere"), we obtain the following estimate: $b_2 = 41.22(1/k)^2 b_1$, i.e., on average, one half-axis of the ellipsoid exceeds the other by 1.5 times.

Let us take a closer look at the VR structure. The maximum velocities $v_x$ of each of the sections lie in the plane x = 0 passing through the circular axis of the vortex (Figure 10). In this plane, $|v_x| > |v_0|$ except for the area close to the circular axis. At infinity, $v_x$ tends asymptotically to $v_0$. The maximum values of the $v_r$ velocities are localized on the cylinder $r = D/2$, vanishing at $x = 0$. At infinity, $v_r$ tends to 0.

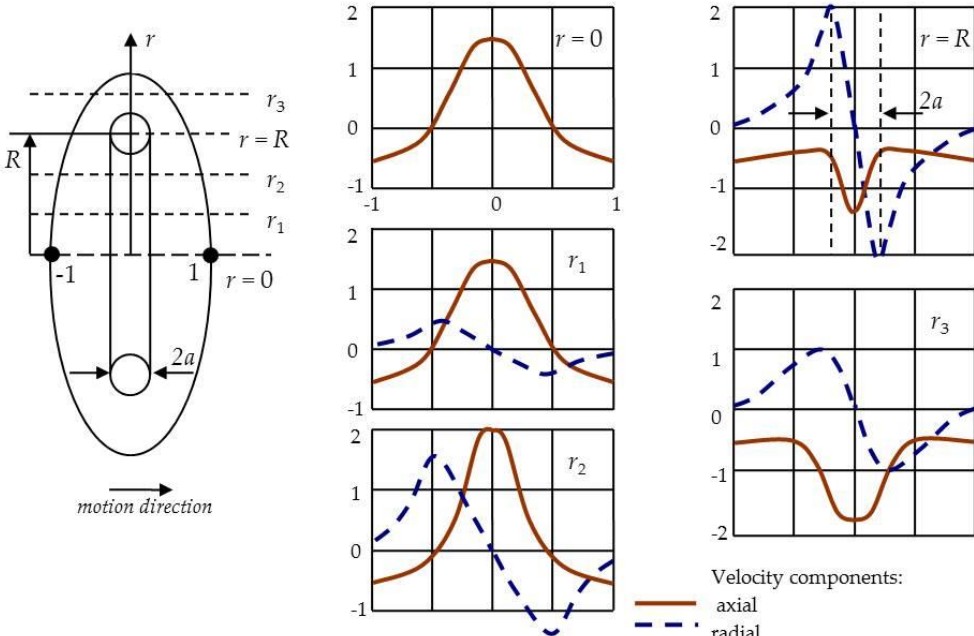

**Figure 10.** Distribution of velocity components over the section of the vortex ring.

Next, we provide a justification for the statement about the possibility of transferring various additives (f. e. smoke particles) by the ring [15]. The velocity in the circulation flow, in contrast to the rotational velocity of a solid, decreases along the hyperbola with increasing radius. The pressure, on the contrary, according to the Bernoulli equation decreases from the periphery to the center.

On the axis of the vortex, the potential flow is disturbed: the vortex core of the circulation flow, having a small diameter, rotates already as a solid body and the Bernoulli equation is not applicable here. In the core, the linear velocity increases from the center to the periphery in proportion to the distance to the boundary between the vortex core and the atmosphere. The position of this boundary depends on the angular velocity of the core.

Using the equation of gas element equilibrium in a steady circulation flow around a single vortex and bearing in mind that in the core $v = \omega r$, where $\omega$ is the vorticity ($s^{-1}$), we obtain the following:

$$\Delta p = \frac{\rho v^2}{r} \Delta r = \rho \omega^2 r \Delta r.$$

The sign of $\Delta p$ coincides with the sign of $\Delta r$, therefore, inside the vortex core, the pressure continues to decrease from the periphery to the center (although the speed also decreases toward the center). Thus, a strong suction occurs in the core, due to which outer particles are drawn (sucked) into the core.

The parameters of the vortex torus can be described using the model of a cylindrical vortex with a finite core of a circular cross section of radius $a$, in which the vorticity $\omega$ is constant—the Rankin vortex [16] (Figure 11). Outside the core, the current is assumed to be vortex-free. Such a vortex can be approximated by a continuous distribution of rectilinear vortex filaments in the core. Then, the cross-sectional element of the core d$S$ gives a contribution to the circulation d$\Gamma$ equal, according to Stokes theorem, to d$\Gamma = \omega$d$S$. The circulation along any contour that once covers the entire core of the vortex is $\Gamma = \omega \pi a^2$ = const.

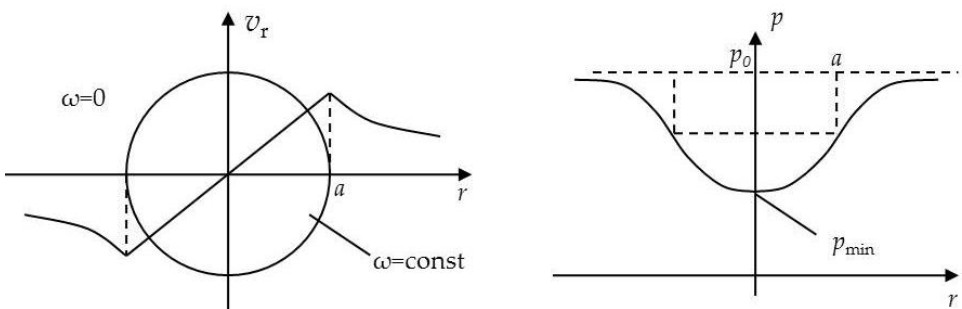

**Figure 11.** Distribution of velocity (**left**) and pressure (**right**) in the Rankin vortex [7].

If we take into account the axial symmetry of the problem, i.e., the presence of only the circumferential component of the velocity $v_r = v_r(r)$, then from the Stokes theorem for a circle of radius $r > 0$ we have $2\pi r v_r = \omega \pi a^2$, from where, taking into account the expression for circulation, we find an expression for the velocity in the region of the vortex-free (potential) flow.

$$v_r = \frac{a^2 \omega}{2r} = \frac{\Gamma}{2\pi r}, r > a.$$

Inside the core, we similarly obtain $2\pi r v_r = \pi r^2 \omega$ or

$$v_r = \frac{\Gamma r}{2\pi a^2}, r < a.$$

The average velocity in the VR core can be estimated as the ratio of the linear velocity at the boundary of the core to the radius of the core. From experimental data [8], at the Reynolds number Re = $7.5 \times 10^4$, the angular velocity of rotation in the core reaches 35,000 rpm. The forward speed of the ring is 11 m/s. In experiments conducted with

the help of a shock tube, rings with a translational velocity of more than 100 m/s were obtained (in particular, at the speed of the outgoing jet of 198 m/s and the radius of the generator nozzle of 25 mm, a ring with a radius of 33 mm moving at a speed of 99 m/s was obtained).

The radial distribution of static pressure is characterized by its sharp decrease in the vortex core. To calculate the pressure profile, we apply the Euler equation, which in polar coordinates, taking into account the axial symmetry, will take the form $\rho \frac{v_\tau^2}{r} = \frac{dp}{dr}$, from where $p = p_\infty + \int_\infty^r \frac{v_\tau^2}{r} dr$, where $\rho$ is the density of the medium; $p_\infty$ the pressure at infinity. Substituting the velocity profiles in the expression for pressure, we find the pressure distribution

$$p = p_\infty - \rho \frac{\omega^2 a^4}{8r^2}, r > a,$$

$$p = p_\infty - \rho \frac{\omega^2 a^2}{4} + \rho \frac{\omega^2 r^2}{8}, r < a$$

The minimum pressure is achieved on the axis of the vortex as follows:

$$p_{\min} = p_\infty - \rho \frac{\omega^2 a^2}{4} = p_\infty - \rho \frac{\Gamma^2}{4\pi^2 a^2}.$$

At the core boundary,

$$p - p_{\min} = \frac{p_{\min} - p_\infty}{2}.$$

For an infinitely thin vortex thread of intensity $\Gamma$, the pressure on the axis tends to $(-\infty)$, as can be seen with $a$ tending to 0.

To determine the initial geometric parameters of the vortex ring, we can write a closed system of equations as follows:

$$\begin{aligned} R_0 &= k_1 D_n / 2 \\ a_0 &= R_0 / k \\ b_{10} &= R_0 / 0.9 \\ b_{02} &= 41.22 \left(1/k^2\right) b_{10} \end{aligned} \tag{1}$$

where $D_n$ is the exit diameter of the nozzle, $R_0$ is the initial radius of the ring, $a_0$ is the initial radius of the ring core; $b_1$ and $b_2$ are the dimensions of the sides of the "atmosphere" of the vortex (taken as an ellipsoid), $k_1 = 1.1 \ldots 1.3$ experimental coefficient that determines the expansion of the ring at the exit from the generator nozzle (when the ring is formed inside the generator, $k_1$ can be taken equal to 1), and $k$ is the coefficient.

Since the vortex is a stable structure, we assume that its geometric parameters change proportionally on the trajectory.

To determine the kinematic characteristics of a vortex ring, we consider a physical model of its motion in the environment.

For further discussion, we will take the following notation: $U$ is the forward velocity of the vortex ring; $u$ is the rate of expansion of the vortex ring along the $y$ axis; $v_{el}$ is the velocity modulus of the vortex element; $\rho_1$ is the density of the medium outside the ring (since the motion of the ring at the earth's surface is considered, we can take $\rho_1 = 1.29$ kg/m$^3$—the density of air under normal conditions); $\rho$ is the density of the medium inside the ring (determined by the density of the "atmosphere"); $\Gamma$ is the velocity circulation; $S$ is the area of the midsection of the vortex; $M$ is the mass of gas in the vorticity region; $M_1$ is the mass of gas in the "atmosphere".

In the framework of the physical model of the process used, based on [17], we assume the following assumptions:

- The gas heated to a certain temperature and located in the generator chamber exits the nozzle in a uniform laminar flow, the core of the vortex is completely formed from this gas, and the "atmosphere" of the vortex is formed from the surrounding cold air;
- During the movement of the vortex, there is a constant mass exchange of the gas forming the "atmosphere" of the vortex with the external environment due to turbulent mixing;
- The air in the core of the vortex does not flow out; therefore, as the size of the core increases, colder air from the "atmosphere" flows into it, which leads to a gradual cooling of the core.

When the ring moves, the following three forces act on it:

(3)  The power of Zhukovsky

$$F_{Zh} = \rho_1 v_{el} \Gamma R d\varphi,$$

The Zhukovsky force acts perpendicular to the gas element motion direction; its vertical component stretches the ring, and the horizontal component slows down its movement (for a circular vortex ring $F_{Zh} = 2\pi\rho_1\Gamma v_{el}R$);

(4)  Hydrodynamic drag force

$$F_{gd} = C_x \rho_1 U^2 S / 2,$$

where $C_x$ is the coefficient of resistance of the vortex, and $S$ is the area of its midsection. The force due to the pressure difference in front of and behind the ring and the viscous friction on the surface of the body, determined by the disruption of the flow from the back side (friction at the boundary, as already noted, is small) and directed opposite to the forward velocity of the ring;

(5)  The Archimedean force, which arises from the difference in the densities of the air inside the ring and in the atmosphere,

$$F_A = (\rho - \rho_1)g V_{rot},$$

where $V_{rot}$ is the volume of the "atmosphere" of the vortex.

Thus, it is possible to obtain the equations of ring motion in projections on orthogonal coordinate axes as follows:

$$\frac{d}{dt}[U(M + M_1)] = -2\pi\rho_1\Gamma Ru - \frac{\rho_1 U v_{el}}{2}C_x S$$
$$\frac{d}{dt}[u(M + M_1)] = -2\pi\rho_1\Gamma RU - \frac{\rho_1 u v_{el}}{2}C_x S + g V_{rot}(\rho_1 - \rho) \tag{2}$$

Analyzing the large array of experimental data obtained, we can conclude that the most acceptable form of estimating the change in the radius of the vortex ring on the trajectory is the formula

$$R(t) = R_0 + \alpha L(t) \tag{3}$$

where $\alpha = R/L$ is the expansion coefficient determined from the experiment (for estimated calculations, we can take $\alpha \approx 0.01$). Thus, the rate of expansion of the ring is constant over the entire trajectory of its motion, and in System (2), we can take $u = \alpha U$, and then consider only the first equation of the system, in which the approximation $U \approx v_{el}$ will be valid.

When decomposing the left part of the equation, we can distinguish the mass transfer factor $vdm/dt$, where $m = 1.08M_1$ is the total mass of the vortex ring, taken from the condition $M_1/M = 12$. Let us assume that mass exchange with the environment occurs only in the "atmosphere," and in the core of the ring, the environment remains unchanged during movement. Estimates for the mass transfer factor $dm/dt$ are semiempirical. Considering the motion of the vortex ring in the form of an ellipse of rotation with a major semiaxis $b_1 = R/0.9$ and with a mass $m = 1.08M_1 = 1.08\pi R_{cg}S_{ell}$, where $R_{cg}$ is the coordinate of the center of gravity of the half of the ellipse, and $S_{ell} = \pi b_1 b_2$, and taking the postulate of

preserving the shape of the vortex, as well as the linear dependence of the increasing radius on the distance traveled, we obtain the ratio

$$\frac{dm}{dt} = \frac{781.5}{k^2} \rho \alpha R^2 U \tag{4}$$

Thus, the mass of the turbulent vortex ring increases linearly in accordance with the distance traveled by the vortex, due to the involvement of media from the environment in the vortex motion. As a result of this process, the kinetic energy is lost, and the vortex is decelerated.

The relationship between the density of the gas forming the vortex and the temperature can be determined using the Clapeyron–Mendeleev equation.

$$\rho T = inv. \tag{5}$$

The temperature of the vortex ring at the current time can be defined as

$$T = \frac{V_0}{V}(T_0 - T_{inv}) + T_{inv} \tag{6}$$

where $V_0$ is the initial volume of the vortex ring, which can be assumed to be equal to the volume of the generator chamber; $T_0$ is the initial temperature of the gas in the ring; $T_{inv}$ is the ambient temperature.

The circulation of the medium in the vortex ring can be determined by the Burgers formula

$$\Gamma = 1.39 \times 2\pi a U \tag{7}$$

Thus, from the first equation of the System (1), we finally obtain the differential equation

$$\frac{dU}{dt} = -\frac{U^2}{m}(A + B + C) \tag{8}$$

where $A = 70.8 \frac{\alpha}{k} R^2$—coefficient of geometric dimensions of the ring; $B = \frac{\rho_1 C_x S}{2}$—coefficient of the geometric shape of the ring; $C = \frac{781.5}{k^2} \rho \alpha R^2$—mass transfer coefficient.

The visualization technique considered in this paper may be used to verify the developed model of the motion of the vortex ring, which is the subject of a separate study. Here, for example, we will present a specific result of comparing theoretical representations with experimental results.

Figure 12 shows a comparison of the velocity calculated from (8) with the experimental data. A rather large spread of experimental data is explained by the difficulty of accurately selecting a certain point of the ring for "tracking" the speed.

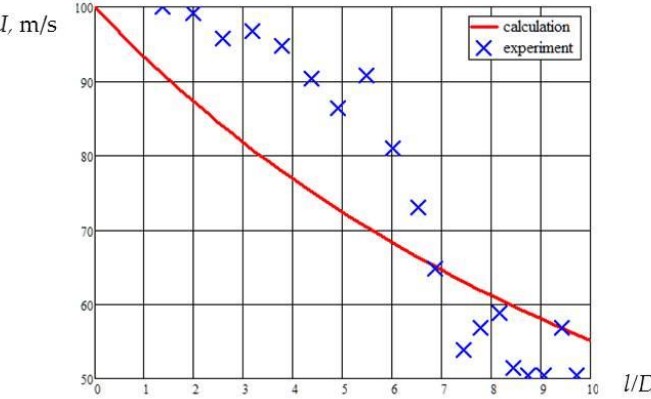

**Figure 12.** Comparison of the calculated dependence of the absolute velocity of the vortex ring on the relative distance with experimental data.

## 4. Conclusions

The experimental data illustrate the possibilities of the presented author's technical solutions for use in the study of ballistics, mechanics, and dynamics of gaseous media. Their use increases the efficiency of scientific research in the field of fundamental and applied research in the field of physics of fast processes.

A series of frames from video files of the physical phenomena considered in the work are presented. The possibility of studying the mechanisms of transition from combustion to detonation using an innovative optical method for recording fast processes is shown. The dynamics and stages of the phenomenon of detonation transfer through a certain size of the gap filled with various rheological media (air, water) are demonstrated. The picture of the cavitation process at the stage of the detonation wave exit to the water surface is revealed and detailed. A variant of verifying the solution to the problem of forming a vortex ring using the results of visualizing the physical process of forming a ring is demonstrated.

The paper demonstrates new possibilities for analyzing the dynamics of gaseous media due to the detailed registration of the visual picture of the ballistic process. The novelty of the presented visualization technology manifests itself when watching video files. In this case, the visual features of the dynamics and stages of the observed phenomena are more clearly distinguished. The results can be easily correlated with computer visualization data. In the last five years, based on the use of the technology of visualization of ballistic processes, with the participation of the authors, a number of research works were carried out using detailed video recordings of the realized physical phenomena. As a result of the completion of the research cycle, the phenomenology of the phenomena was revealed, and the virtual models of these processes were verified. One of the options for studying the process of vortex ring formation is presented in this work. The technology and launcher presented in this work were awarded two silver medals in 2018 at the XXI International Salon of Inventions and Innovative Technologies.

**Author Contributions:** Conceptualization, M.S.; methodology, V.S. and D.L.; software, D.L.; validation, M.S. and D.L.; formal analysis, V.S.; investigation, M.S.; resources, V.S.; data curation, D.L.; writing—original draft preparation, M.S.; writing—review and editing, M.S.; visualization, D.L.; supervision, M.S.; project administration, V.S.; funding acquisition, V.S. All authors have read and agreed to the published version of the manuscript.

**Funding:** This research received no external funding.

**Institutional Review Board Statement:** Not applicable.

**Informed Consent Statement:** Not applicable.

**Data Availability Statement:** Not applicable.

**Acknowledgments:** The authors are grateful to the staff of the Department "High-Precision Airborne Devices" of the Bauman Moscow State Technical University and D.V. Gelin, D.A. Lysov, V.A. Markov, and A.V. Petyukov for their active participation in the experiments and for the data provided.

**Conflicts of Interest:** The authors declare no conflict of interest.

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
