# Peer review of "Special Cases of Using Visualization Technology for Analyzing the Dynamics of Gaseous Environment"

_fluids, doi:10.3390/fluids6080290_

Round 1
Reviewer 1 Report
I don’t see the professional excellence of the article. The authors present the visualization technique used in the experiment. The theoretical contexts prescribed are part of the university curriculum.
And there are also problems in presenting the article:
- The authors refer to several (5 in total) tables, but I did not find any tables in the manuscript.
- There are different forms of citations for the figures (Figure, Fig., fig.)
Reviewer 2 Report
The paper reports about a new measurement and visualization technique used to unveils very fast gas-dynamic processes as detonation and wave propagation and the development of a vortex torus exiting from a tube. So, the paper is rather a patch work of two topics than a monograph. From my point of view, it is not beneficial for single paper.
Although the sentences are almost grammatically correct, the structure of some sentences does not follow the typical English style. A rework of the first part of the paper enhances the readability significant.
In some respect I’ve got the feeling the paper is incomplete. After a very detailed and good description of the physical and mathematical properties of the vortex ring (Line 351 till 473) I expected a discussion of the findings in form of diagrams etc. (e.g., pmin (line 376) of the observed vortex ring (line 363-366)). But the paper ends with a single diagram (fig 12).
I recommend a significant rework before publication. The correctness of the equations in the draft of a paper was not checked by myself.
Additional comments:
Line 87: What is meant by CINE format? To my knowledge CINE is a proprietary computer file format containing video date and not a visualization method. Is meant here a visualization frame by frame or slow motion or a special kind of frame postprocessing?
Line 112/160 Phantom instead of Fantom.
Line 102 Table 2 is missing
Line 119-121 are exactly the same as Line 116-118
Line 166ff A sketch or notification inside the pictures of the active and passive volume and air gap is strongly recommended.
Line 479 Table 5 is missing and “conclusions” is obviously the header of the next section.
Reviewer 3 Report
This study is related to a new optical visualization technology which was used in applied research when observing and modeling the dynamics of the flow of gaseous environment. This paper describes well the visualization results those the engineers and reserachers are interested in.
Round 2
Reviewer 1 Report
I still maintain my opinion that the scientific level of this article does not reach the level to publish in the journal Fluids.
Reviewer 2 Report
The paper was slightly improved but it is still far below the potential the new technique can provide. It concerns the image processing methods applicable (RAW image format) as well as the strong link between theoretical derivations (2nd part of the paper) and the observed/recorded phenomena (1st part).
From my point of view, it does not increase the reputation of the researchers. Nevertheless, I accept the paper in the reviewed version for publication.